# Electronic Floquet gyro-liquid crystal

Iliya Esin [1,2✉], Gaurav Kumar Gupta[1], Erez Berg [3], Mark S. Rudner [4] & Netanel H. Lindner [1]

Floquet engineering uses coherent time-periodic drives to realize designer band structures on-demand, thus yielding a versatile approach for inducing a wide range of exotic quantum many-body phenomena. Here we show how this approach can be used to induce non-equilibrium correlated states with spontaneously broken symmetry in lightly doped semiconductors. In the presence of a resonant driving field, the system spontaneously develops quantum liquid crystalline order featuring strong anisotropy whose directionality rotates as a function of time. The phase transition occurs in the steady state of the system achieved due to the interplay between the coherent external drive, electron-electron interactions, and dissipative processes arising from the coupling to phonons and the electromagnetic environment. We obtain the phase diagram of the system using numerical calculations that match predictions obtained from a phenomenological treatment and discuss the conditions on the system and the external drive under which spontaneous symmetry breaking occurs. Our results demonstrate that coherent driving can be used to induce non-equilibrium quantum phases of matter with dynamical broken symmetry.

[1] Physics Department, Technion, Haifa, Israel. [2] Department of Physics, California Institute of Technology, Pasadena, CA, USA. [3] Department of Condensed Matter Physics, Weizmann Institute of Science, Rehovot, Israel. [4] Center for Quantum Devices and Niels Bohr International Academy, Niels Bohr Institute, University of Copenhagen, Copenhagen, Denmark. ✉email: iesin@caltech.edu

Quantum systems that are driven far from thermal equilibrium have been shown to support a wide variety of exotic phases of matter, with properties that have no analogues in equilibrium. Examples include quantized charge pumping[1–4], spatio-temporal symmetry breaking (as in time crystals)[5–11], and topological phases with unconventional types of edge states that defy the equilibrium bulk-edge correspondence[12–23]. Realizing many of these examples requires highly engineered setups and strong isolation from the environment. Here, we find an intrinsically non-equilibrium phase that can be naturally realized in a steady state of an optically driven semiconductor. The transition to this phase is signaled by a unique combination of a change of topology of the Fermi surface, accompanied by a rotating orientational (ferromagnetic-nematic) order parameter.

From a mechanistic point of view, in equilibrium, strongly correlated electronic phases emerge from the competition between the potential energy savings and kinetic energy costs of developing correlations that allow electrons to avoid each other. In materials with band structures that feature large densities of states (DOSs), the kinetic energy costs that oppose the formation of correlations are small. Such materials, therefore, provide a rich platform for realizing exotic phases of matter where interparticle interactions crucially alter the ground-state properties of the system. Since the concept of a ground state does not apply to driven quantum systems, it is an interesting question whether an analogous mechanism can lead to strongly correlated non-equilibrium phases.

Here, we address this question in the context of two-dimensional (2D) electron systems. A prominent route to achieving high DOS bands in 2D is through the application of strong out-of-plane magnetic fields, which gives rise to flat Landau levels. At certain rational filling fractions, the resulting macroscopic degeneracy is lifted by the formation of strongly correlated fractional quantum Hall states[24,25]. Recently, a rich phase diagram of correlated states arising from flat band formation has also been uncovered for twisted bilayer graphene, when the twist angle between layers is tuned close to the "magic angle"[26–31].

Two-dimensional systems in which the minimum of the single-particle dispersion occurs along a ring in momentum space (rather than at a single point, as for a standard parabolic dispersion), provide an alternative route for achieving large DOSs and exotic correlated phases[32–45]. This occurs, for example, in two-dimensional materials with strong Rashba-type spin-orbit coupling[46,47]. The ring-minimum in such systems leads to a large degeneracy and a divergent DOS at energies approaching the bottom of the band. At low densities, inter-particle interactions may lead to a plethora of possible symmetry-broken phases. In particular, for short-ranged interactions, electronic liquid-crystalline ground states were predicted in ref. [39]. These phases exhibit spontaneously broken rotational symmetry, with highly anisotropic Fermi surfaces and related susceptibilities.

Here, we present a "Floquet engineering"[48–77] approach for inducing non-equilibrium liquid crystalline phases by subjecting 2D electron systems to optical driving fields. The non-equilibrium phase transition that we describe results from an interplay between coherent driving, electron–electron interactions, and dissipative dynamics due to the system's coupling to its environment[78–92]. The coherent drive is used to produce a Floquet band structure that features a ring-like minimum analogous to that of the Rashba system described above. In turn, the interactions and dissipative dynamics determine the steady state of the system and the symmetry breaking that it exhibits. Intriguingly, we find that the system spontaneously develops strong anisotropy, with a directionality that rotates periodically in time.

We refer to this exotic order as *gyro-ferromagnetic nematic* (GFN) order.

## Results

**Physical mechanism and theoretical approach.** A ring-like dispersion minimum is natural to obtain in a direct band gap system subjected to a coherent drive, where the drive frequency $\Omega$ is larger than the system's band gap (Fig. 1a). The structure of the modified (Floquet) bands is most easily visualized in a rotating frame. Starting from the original bands as depicted in Fig. 1a, we transform to a rotating frame in which the energies of all states in the valence band are rigidly shifted upwards by $\hbar\Omega$[67]. In the rotating frame, the (shifted) valence and conduction bands cross along a continuous "resonance ring" of points in momentum space where the original conduction and valence bands were separated by $\hbar\Omega$ (see green curves in Fig. 1a). After transforming to the rotating frame, the driving field obtains a static (co-rotating) part, and a component that oscillates with integer multiples of the drive frequency $\Omega$. Within the rotating wave approximation we keep only the static part of the drive in the rotating frame, and discard the oscillating components. As we show in detail below, under appropriate conditions on the material's band structure and the form of the drive, the co-rotating part of the drive opens a "Floquet gap" all the way around the resonance ring. The minima and maxima of the resulting upper and lower Floquet bands correspondingly occur along a ring in momentum space (Fig. 1b), yielding a DOS for the Floquet bands, $D_F(\varepsilon)$, with square-root divergences near the two-band extrema (Fig. 1d). Along these

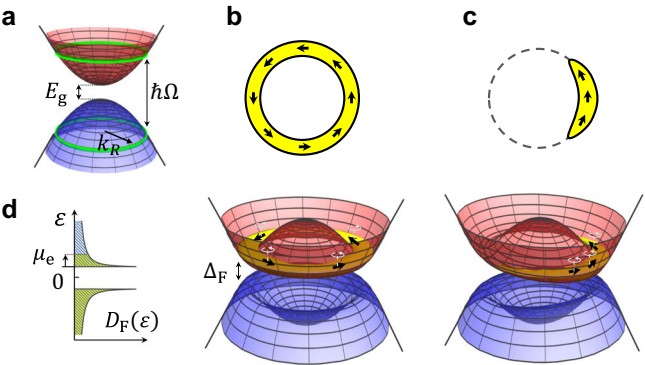

**Fig. 1 Floquet band structure near the Γ-point. a** The band structure of the non-driven semiconductor. The resonance rings of the external drive are indicated by the green curves. **b, c** Floquet quasienergy bands arising from the semiconductor's band structure and the resonant drive around $\varepsilon = 0$. The yellow area represents the occupation of the upper Floquet band in the "ideal" distribution scenario, analogous to the zero-temperature Gibbs state for the quasi-energy spectrum. Black arrows represent the pseudospin direction of the Floquet states near the resonance ring. The texture of the pseudospins arises from the pseudospin-momentum locking induced by the semiconductor. In addition, each pseudospin rotates in the x-y plane with the frequency of the periodic drive as is indicated on the figure by the light-gray thin arrows attached to each pseudospin. In the symmetric phase, **b** due to rotational symmetry, the Floquet states near the resonance ring are uniformly occupied, as is indicated below panel **b**. Panel **c** demonstrates the single-particle Floquet bands in the broken symmetry phase. In this case, the resonance ring is tilted towards a spontaneously chosen direction. The occupation of the bands is then biased toward this direction, signaling a ferromagnetic alignment of the pseudospins. **d** The density of Floquet states as a function of the quasi-energy around $\varepsilon = 0$ in the paramagnetic phase. The density of states features square-root Van Hove singularities in each Floquet band, i.e., $D_F(\delta\varepsilon) \sim \delta\varepsilon^{-1/2}$ in the upper Floquet band, where $\delta\varepsilon \equiv \varepsilon - \Delta_F/2$. A similar relation holds for the lower Floquet band.

ring-extrema, the Floquet-Bloch states may be characterized by a pseudo-spin with a non-trivial winding, see Fig. 1b, in close analogy to the spin winding that occurs around the ring minimum of a Rashba-type band structure.

Our goal is to find the conditions under which the divergence in the DOS promotes spontaneous symmetry breaking in the electronic steady-state of the system. Throughout this paper we study a system lightly doped above half filling. To preview the considerations involved, consider first an ideal situation where the steady-state is a zero-temperature Gibbs state of electrons in the band structure obtained in the rotating wave approximation[93]. In the absence of electron–electron interactions this zero temperature state corresponds to a full lower Floquet band and an annulus-shaped Fermi sea at the bottom of the upper band, as indicated in Fig. 1b. At low-doping density, the DOS at the Fermi energy is strongly enhanced due to the divergence at the band bottom. Sufficiently strong electron–electron interactions make it energetically favorable for electrons to change the topology of the Fermi sea from an annulus to a single pocket centered around a spontaneously chosen point on the resonance ring. The symmetry broken state is ferromagnetic as the pseudospins of the electrons are predominantly aligned along one direction, see Fig. 1c. This leads to a reduction in the potential energy (as the electronic wavefunction overlaps are suppressed for parallel pseudospins due to Pauli exclusion). As a consequence the mean-field band structure in the symmetry broken phase acquires a tilt along the spontaneously chosen momentum direction. Interestingly, due to the periodic time-dependence of the Floquet states, the emergent pseudospin magnetization vector rotates in time at the frequency of the drive (in the lab frame), giving rise to the GFN order.

The discussion above, based on the rotating wave approximation and a zero-temperature Gibbs-type steady state, captures the essence of the symmetry-breaking transition. However, the non-equilibrium nature of the setup implies that the steady-state cannot be described by a simple Gibbs state. Even when the electronic system is coupled to a zero-temperature heat bath, unavoidable scattering processes create electron–hole excitations in the Floquet bands. These excitations may suppress the tendency towards ordering. In this work, we introduce a self-consistent treatment of coupled kinetic and Floquet-Hartree-Fock equations that capture the interplay between the steady state of the system and its renormalized Floquet band structure, with the possibility of spontaneously broken symmetry. Using this treatment, we will obtain the non-equilibrium phase diagram for different doping densities, interaction strengths, and properties of the external heat baths.

**Model system and problem setup.** To study the phase transition in the steady-state of a periodically driven direct bandgap semiconductor, we introduce an effective model that describes the single-particle electronic states near the semiconductor's $\Gamma$-point. We consider a two-band, two-dimensional (2D) system, with topologically trivial bands. (This model, which lacks time-reversal symmetry, may be taken to represent half of the degrees of freedom of a time-reversal symmetric semiconductor[67,94].) We assign a common effective mass $m_*$ for the electrons in the conduction band and holes in the valence band, and denote the gap separating these bands by $E_g$, see Fig. 1a. The Hamiltonian describing the electronic system and the time-periodic drive, near the $\Gamma$-point, reads

$$\hat{\mathcal{H}}(t) = \sum_{\boldsymbol{k}} \hat{\boldsymbol{c}}_{\boldsymbol{k}}^{\dagger}[H_0(\boldsymbol{k}) + H_{\mathrm{d}}(t)]\hat{\boldsymbol{c}}_{\boldsymbol{k}} + \sum_{\boldsymbol{q}} \mathcal{V}_{\boldsymbol{q}}\hat{\rho}_{\boldsymbol{q}}\hat{\rho}_{-\boldsymbol{q}}. \quad (1)$$

Here $H_0(\boldsymbol{k}) = E_0 + (\hbar^2|\boldsymbol{k}|^2/2m_* + E_g/2)\sigma^z + \lambda_0\,\boldsymbol{k}\cdot\boldsymbol{\sigma}$, where $\boldsymbol{k} = (k_x, k_y)$ is the two-dimensional momentum, $\hat{\boldsymbol{c}}_{\boldsymbol{k}}^{\dagger} = (\hat{c}_{\boldsymbol{k}\uparrow}^{\dagger}, \hat{c}_{\boldsymbol{k}\downarrow}^{\dagger})$ is the two-component spinor for the pseudospin degree of freedom,

$\mathcal{V}_{\boldsymbol{q}}$ describes an effective short-ranged electron–electron interaction, $\hat{\rho}_{\boldsymbol{q}} = \sum_{\boldsymbol{k}}\hat{c}_{\boldsymbol{k}+\boldsymbol{q}}^{\dagger}\hat{c}_{\boldsymbol{k}}$, and $E_0$ is an energy offset. We denote the pseudospin-orbit coupling by $\lambda_0$ and use $\boldsymbol{\sigma} = (\sigma^x, \sigma^y)$, where $\sigma^{\alpha}$, $\alpha = x, y, z$, is a Pauli matrix in the pseudospin space. The band-structure of the system in the absence of the drive is given by the spectrum of $H_0$. We denote the energies of the valence and conduction band by $E_v(\boldsymbol{k})$ and $E_c(\boldsymbol{k})$, respectively.

We consider a uniform driving field that couples to the electrons through $\sigma^z$, $H_{\mathrm{d}}(t) = V\cos(\Omega t)\sigma^z$, with an above-gap frequency $\hbar\Omega = E_g + \delta E$, where $\delta E$ is much smaller than semiconductor's full bandwidth. More realistic time-dependent electromagnetic fields can be incorporated in this model, see ref. [67]. The Floquet state solutions of $\hat{\mathcal{H}}(t)$ for $\mathcal{V}_{\boldsymbol{q}} = 0$ satisfy $\left[i\hbar\frac{\partial}{\partial t} - H_0(\boldsymbol{k}) - H_{\mathrm{d}}(t)\right]|\psi_{\boldsymbol{k}\nu}(t)\rangle = 0$, with $|\psi_{\boldsymbol{k}\nu}(t)\rangle = e^{-i\varepsilon_{\boldsymbol{k}\nu}t/\hbar}|\phi_{\boldsymbol{k}\nu}(t)\rangle$. Here $|\phi_{\boldsymbol{k}\nu}(t)\rangle = |\phi_{\boldsymbol{k}\nu}(t+\mathcal{T})\rangle$ is periodic with period $\mathcal{T} = 2\pi/\Omega$ and $\varepsilon$ is the quasienergy (which is periodic in $\hbar\Omega$). Throughout, we use the convention $-\hbar\Omega/2 \leq \varepsilon < \hbar\Omega/2$. For convenience we take $E_0 = \hbar\Omega/2$ such that $\varepsilon = 0$ at the center of the Floquet gap (Fig. 1d).

The drive resonantly couples valence and conduction band states along a ring in momentum space for which $\hbar\Omega = E_c(\boldsymbol{k}) - E_v(\boldsymbol{k})$. We denote the radius of this ring by $k_R$. At the resonance ring, a gap of magnitude $\Delta_F = 2\lambda_0 k_R V/\hbar\Omega$ opens in the Floquet quasienergy spectrum. This gap separates the "upper Floquet" ($\nu = +$) and "lower Floquet" ($\nu = -$) bands, corresponding respectively to $\varepsilon > 0$ and $\varepsilon < 0$. Here, we will focus on the parameter regime $\Delta_F \ll \delta E$, where the ring minimum is well developed. Each of the bands has a ring of degenerate states associated with square-root van Hove singularities in the density of Floquet states: near the bottom of the upper Floquet band, the density of states takes the form $D_F(\delta\varepsilon) \approx \frac{m_*}{2\pi\hbar^2}\sqrt{\frac{\Delta_F}{\delta\varepsilon}}$, where $\delta\varepsilon = \varepsilon - \Delta_F/2$, see Fig. 1. A similar expression holds for quasienergies near the top of the lower Floquet band. Below we show how these van Hove singularities promote spontaneous symmetry breaking in the driven system.

**Order parameter and Floquet mean field approach.** In this work we will look for spontaneous symmetry breaking that emerges in the steady state of the driven system. The steady state arises from an interplay between the time-periodic drive, electron–electron interactions, and the coupling of the electrons to the electromagnetic and phononic modes of their environment. In this interplay, the electron–electron interactions play a dual role, as they lead to formation of order parameters as well as to incoherent scattering which may suppress the tendency towards order.

In order to capture the coherent part of the electron–electron interaction, which leads to order parameter formation, we use a mean-field approximation in which we assume that the steady state is Gaussian (i.e., obeys Wick's theorem). We assume translation invariance is maintained, and consider a mean-field decoupling of the Hamiltonian Eq. (1) with ferromagnetic nematic order parameter

$$\boldsymbol{h}_{\boldsymbol{k}}(t) = -\sum_{\boldsymbol{k}'} \mathcal{V}_{\boldsymbol{k}-\boldsymbol{k}'}\langle\hat{\boldsymbol{c}}_{\boldsymbol{k}'}^{\dagger}\boldsymbol{\sigma}\hat{\boldsymbol{c}}_{\boldsymbol{k}'}\rangle_{\mathrm{MF}}. \quad (2)$$

The expectation value in Eq. (2) is taken with respect to the time-periodic steady-state of the system. The corresponding mean-field Hamiltonian is given by $\hat{\mathcal{H}}_{\mathrm{MF}}(t) = \sum_{\boldsymbol{k}}\hat{\boldsymbol{c}}_{\boldsymbol{k}}^{\dagger}H_{\mathrm{MF}}(\boldsymbol{k}, t)\hat{\boldsymbol{c}}_{\boldsymbol{k}}$, where

$$H_{\mathrm{MF}}(\boldsymbol{k}, t) = H_0(\boldsymbol{k}) + H_{\mathrm{d}}(t) + \boldsymbol{h}_{\boldsymbol{k}}(t)\cdot\boldsymbol{\sigma}. \quad (3)$$

Note that if $\boldsymbol{h}_{\boldsymbol{k}}(t)$ has the same time-period as the drive, $\hat{\mathcal{H}}_{\mathrm{MF}}(t)$ is also time-periodic and therefore defines a new Floquet problem.

The time-periodic steady state used in Eqs. (2) and (3) is determined self-consistently by solving the kinetic equation for

the populations of electrons in the Floquet bands of $\hat{\mathcal{H}}_{\mathrm{MF}}(t)$. These populations are defined as $f_{k\nu}(t) \equiv \langle \hat{\phi}_{k\nu}^{\dagger}(t) \hat{\phi}_{k\nu}(t) \rangle$, where $\hat{\phi}_{k\nu}^{\dagger}(t)$ is a creation operator corresponding to the Floquet state $|\phi_{k\nu}(t)\rangle$. Note that the meaning of the index $\nu$ and the values of populations $f_{k\nu}$ depend on the order parameter, $h_k(t)$, as it determines the Floquet bandstructure of $H_{\mathrm{MF}}(k, t)$. The kinetic equation includes scattering rates due to electron–phonon interactions, $I_{k\nu}^{s}$, radiative recombination, $I_{k\nu}^{\ell}$, and electron–electron collisions, $I_{k\nu}^{ee}$, and is given by

$$\dot{f}_{k\nu} = I_{k\nu}^{s}(\{f\}) + I_{k\nu}^{\ell}(\{f\}) + I_{k\nu}^{ee}(\{f\}), \qquad (4)$$

where the steady state is determined by $\dot{f}_{k\nu} = 0$. The notation $\{f\}$ refers to the full set of populations over all momenta and band indices.

In writing the kinetic equation in terms of the populations $f_{k\nu}$ we have assumed that the Gaussian steady state is approximately described by a single-particle density matrix which is diagonal in the Floquet basis. This condition is satisfied when the scattering rates in the steady state are small, $\hbar/(\tau_{\mathrm{scat}} \Delta_{\mathrm{F}}) \ll 1$[82]. Here $1/\tau_{\mathrm{scat}}$ is the total scattering rate of the electrons.

The scattering rates $I_{k\nu}^{s}$ and $I_{k\nu}^{\ell}$ describe scattering processes in which a boson (phonon, s, or photon, $\ell$) is emitted or absorbed by the electronic system. The corresponding rates are determined by the dispersions of these bosons, and the form of the electron-boson coupling.

We denote by $\hat{b}_{pq}^{\dagger}$ the operator creating an acoustic phonon (for $p = s$) or a photon (for $p = \ell$) with the three-dimensional (3D) momentum $q = (q_{\parallel}, q_z)$ and frequency $\omega_q = v_p|q|$. Here $q_{\parallel}$ is the component of $q$ within the plane of the 2D electronic system and $v_s(v_{\ell})$ is the speed of sound (light). Note that the phonons propagate in the 3D substrate of the 2D electronic system.

The electron-boson coupling is described by the Hamiltonian[95]

$$\hat{\mathcal{H}}_{\mathrm{HB}} = \sum_{k,p,q} \hat{c}_k^{\dagger} \mathcal{M}_p(q_{\parallel}, \omega_q) \hat{c}_{k+q_{\parallel}} (\hat{b}_{p,q}^{\dagger} + \hat{b}_{p,-q}) + \mathrm{h.c.}, \qquad (5)$$

where $\mathcal{M}_p(q_{\parallel}, \omega_q)$ is the coupling matrix in pseudospin space. We consider a diagonal electron–phonon coupling matrix in the $\{\uparrow, \downarrow\}$ basis, which captures the conservation of the pseudospin in small-momentum-transfer electron–phonon interactions. In contrast, photon emission requires changing the electronic angular momentum. We account for this by taking an electron-photon coupling matrix that is strictly off-diagonal in the $\{\uparrow, \downarrow\}$ basis, as these two basis states have opposite parity. Throughout the manuscript, we will assume that the phonons and photons are in thermodynamic equilibrium at zero temperature.

The rates $I_{k\nu}^{s}$ and $I_{k\nu}^{\ell}$ in Eq. (4) can be computed through Floquet-Fermi's golden rule[96] using the electron-boson coupling in Eq. (5). Similarly, $I_{k\nu}^{ee}$ is computed using Floquet-Fermi's golden rule and the electron–electron interactions appearing in Eq. (1). Explicit expressions for these rates appear in Supplementary Eq. (120).

We argue that despite the non-equilibrium nature of the system we study, the application of a mean-field treatment of interactions can be justified at a similar level as for an equilibrium system. To this end, we identify a limit in which the system we study maps to an equilibrium system. This limit is realized when the gap of the semiconductor, and hence the driving frequency, is large compared to the Rabi frequency (driving amplitude). In this situation, and for the moment neglecting electron–hole radiative recombination, one can apply the rotating wave approximation to the full many-body dynamics including the system-bath coupling. In the rotating frame the system then exactly maps to an equilibrium problem with a static Hamiltonian describing interacting electrons with a new bandstructure and system-bath

couplings[82,93]. Therefore, the steady state of the system would be a Gibbs state with respect to this static Hamiltonian. Here, the mean-field approach can be used to study the properties of this Gibbs state and map out a phase diagram, with the same level of justification as in equilibrium.

Away from the above limit, the many-body dynamics involves additional processes which are not present in equilibrium and which lead to a deviation from the exact Gibbs state discussed above. These processes are often called Floquet-Umklapp processes (see below), since the total initial quasi-energy of the electrons and excitations in the environment differs from the final value by an integer multiple of the driving frequency. The rates for these processes are suppressed in powers of the ratio of the driving amplitude to the driving frequency (in addition, radiative recombination is naturally a slow process relative to electron–phonon and electron–electron scattering), see "Discussion" and Supplementary Note 3d. Therefore, in the system we study, the rates for Floquet-Umklapp processes are small compared with the rates for processes that relax the system to the Gibbs state exhibited by the system in their absence. Our expectation is that the steady state and the order parameter that it exhibits evolve smoothly as the rates of these processes are increased from zero. Therefore, the mean-field approach remains a good approach for studying symmetry breaking even when Floquet-Umklapp processes are present.

**GFN steady states**. Before presenting the full steady-state solution to Eqs. (2), (3), and (4), we introduce a phenomenological model which we will use to characterize the phase diagram of the system. The model includes the key processes required for obtaining the steady-state distribution for the electrons. Our goal is to identify the conditions on the electronic system and its environment under which spontaneous symmetry breaking may occur. A key quantity for describing the steady state is the density of electrons in the upper Floquet band, defined as $n_e = \int \frac{d^2k}{(2\pi)^2} f_{k+}$. Likewise, the density of holes in the lower band, $n_h$, is computed by integration over $1 - f_{k-}$. In what follows, we discuss the generation and annihilation rates of electron–hole pairs (in the Floquet basis) resulting from collision processes [see Eq. (4)]. We refer to these as heating and cooling processes, respectively. Of particular importance are Floquet-Umklapp processes, in which the energies of the electrons and bosonic modes in the initial and final states differ by $\hbar\Omega$. At zero bath temperature, these processes provide the only mechanism for heating.

We will be interested in the situation in which the system is doped slightly above half filling. In the absence of Floquet-Umklapp processes and at zero bath temperature, the steady-state is a zero-temperature Gibbs distribution of electrons in the (mean-field) Floquet bands[93]. Specifically, in this situation, the steady-state features a completely filled lower Floquet band, and a low-density Fermi sea of electrons in the upper Floquet band. In the presence of Floquet-Umklapp processes, this ideal distribution is perturbed by the creation of (inter-Floquet-band) electron–hole pairs. We will focus on the regime where the densities of electrons and holes in the upper and lower Floquet bands are low: $n_e, n_h \ll \mathcal{A}_R$, where $\mathcal{A}_R \equiv \pi k_R^2$ is the area in reciprocal space enclosed by the resonance ring.

The pair creation rate in almost empty upper and almost full lower Floquet bands is approximately independent of the densities of electrons and holes in the respective bands. We denote the total pair creation rate due to collisions with both phonons and photons by $\dot{n}_e|_{\mathrm{ph}} = \Gamma_{\mathrm{ph}}$. Similarly, the pair creation rate due to electron–electron collisions is denoted by $\dot{n}_e|_{ee} = \Gamma_{ee}$. The parameter $\Gamma_{ee}$ depends on $\mathcal{V}_q^2$ at $q$ corresponding to the

inverse interparticle distance in the nearly filled band. The processes contributing to $\dot{n}_e|_{ee}$ are of the Floquet-Umklapp type, and are suppressed by $(V/\hbar\Omega)^2$. In addition, electron–electron scattering gives rise to quasienergy conserving processes, causing thermalization of the populations within each band without changing the electron and hole population densities. These processes therefore do not contribute to $\dot{n}_e|_{ee}$. Moreover, as in equilibrium, these elastic scattering processes all together preserve the form of the distribution when the electrons are distributed according to the Fermi-function over the quasienergy spectrum.

Once excited, the electrons (holes) rapidly relax to the bottom (top) of the Floquet band through multiple low-energy phonon emissions. The electron–hole pairs then annihilate through inter-Floquet-band scattering processes mediated by phonons. The electron–hole pair annihilation processes predominantly occur near the resonance ring, where the electrons and holes are concentrated. Note that for these momenta the Floquet states are equal superpositions of the conduction and valence bands [see Eq. (9)], and these states are efficiently coupled by acoustic phonons. The rate of the pair annihilation processes, $\dot{n}_e|_{cool}$, is proportional to the product of the densities of electrons and holes. Therefore, we estimate $\dot{n}_e|_{cool} = -\Lambda_{inter} n_e n_h$, where $\Lambda_{inter}$ is independent of the populations. Note that for this essential cooling process to occur, the Debye frequency of the phonons needs to be larger than the Floquet gap $\Delta_F$.

Summing up the cooling and heating rates we obtain a rate equation for the density of electrons in the upper Floquet band,

$$\dot{n}_e = \Gamma_{ph} + \Gamma_{ee} - \Lambda_{inter} n_e n_h. \qquad (6)$$

In the steady-state ($\dot{n}_e = 0$), Eq. (6) leads to $n_e n_h = \kappa$, where we define the "heating parameter" $\kappa \equiv \kappa_{ph} + \kappa_{ee}$, with $\kappa_{ph} \equiv \Gamma_{ph}/\Lambda_{inter}$, $\kappa_{ee} \equiv \Gamma_{ee}/\Lambda_{inter}$. Furthermore, the difference between electron and hole excitation densities is fixed by the electron doping, $\Delta n$, measured relative to half-filling, $n_e - n_h = \Delta n$. Using this relation, together with the steady-state solution to Eq. (6) we obtain

$$n_{e/h} = \sqrt{(\Delta n/2)^2 + \kappa} \pm \Delta n/2, \qquad (7)$$

where the plus (minus) sign on the right hand side corresponds to the density of electrons (holes). Note that in the absence of drive-induced heating processes ($\kappa = 0$), the ideal steady-state with no holes in the lower band and density $\Delta n$ in the upper band is obtained. In what follows, we focus on the electron-doped regime, $\Delta n \geq 0$ (similar considerations apply in the hole-doped regime).

Having established the steady-state densities of electrons and holes (concentrated near the Floquet band extrema at the resonance ring), Eq. (7), we are well-positioned to address the conditions for spontaneous breaking of rotational symmetry in the system. In the following, we assume contact interactions described by a constant in $\boldsymbol{q}$ interaction strength, $\mathcal{V}_{\boldsymbol{q}} = U/\varpi$, and $\boldsymbol{k}$-independent magnetization $\boldsymbol{h}(t) = \boldsymbol{h}_{\boldsymbol{k}}(t)$ [see Eq. (2)], where $\varpi$ is the area of the system. In the steady state, $\boldsymbol{h}(t)$ is time periodic with the same time period as the drive. Therefore, we expand $\boldsymbol{h}(t)$ in terms of its Fourier harmonics,

$$\boldsymbol{h}(t) = \mathrm{Re}\left[\boldsymbol{h}_0 + \boldsymbol{h}_1 e^{i\Omega t} + \cdots\right]. \qquad (8)$$

Here $\boldsymbol{h}_0$ and $\boldsymbol{h}_1$ are vectors of complex magnitudes, representing the constant and the first harmonic components of the mean-field, respectively, and "$\cdots$" represents higher harmonics. The values of the coefficients $\{\boldsymbol{h}_i\}$ are determined self-consistently via Eqs. (2), (3), and (4).

Crucially, a nonvanishing magnitude of the "in-plane" ($x$-$y$) component of the magnetization $\boldsymbol{h}(t)$, which we denote by $\boldsymbol{h}^{(xy)}(t)$, does not respect the rotational symmetry of the microscopic Hamiltonian $\hat{\mathcal{H}}(t)$, see Eq. (1). Therefore, $|\boldsymbol{h}^{(xy)}(t)|$ serves as the

order parameter for the GFN phase that we study. In contrast, a non-vanishing $z$ component of $\boldsymbol{h}(t)$ respects the symmetry. Generically, we expect a non-vanishing $z$ component of $\boldsymbol{h}(t)$ in both the symmetry broken and unbroken phases. In particular, we expect a large static $z$ component of $\boldsymbol{h}_0$ (with magnitude on the order of $U$) even in the absence of the drive. This static field simply renormalizes the parameters of $H_0$ in Eq. (1), and therefore we do not treat it self-consistently in our analysis.

For simplicity, in the analytical treatment below we take $\boldsymbol{h}_n = 0$ for $n \geq 2$ since these harmonics are suppressed by powers of $V/(\hbar\Omega)$ for $V/(\hbar\Omega) \ll 1$. Furthermore, we note that when the in-plane ($x$-$y$) components of $\boldsymbol{h}_0$ are small, $|\boldsymbol{h}_0^{(xy)}| \ll E_g$, their effect on the Floquet band structure via Eq. (3) is negligible. To facilitate the analysis we thus also take $\boldsymbol{h}_0 = 0$, thereby focusing our attention on the behavior of $\boldsymbol{h}_1$, which describes the component of the magnetization that oscillates at the same frequency as the drive. In the next section, we will present numerical results in which all harmonics are allowed to freely develop.

In order to understand the expected form of $\boldsymbol{h}_1$, it is helpful to examine the Floquet states near the resonance ring. These states are created by the operators

$$\hat{\phi}_{\boldsymbol{k}\pm}^{\dagger}(t) = (e^{-i\Omega t}\hat{c}_{\boldsymbol{k}\uparrow}^{\dagger} \mp e^{i\theta_{\boldsymbol{k}}}\hat{c}_{\boldsymbol{k}\downarrow}^{\dagger})/\sqrt{2} + \mathcal{O}(V/\hbar\Omega), \qquad (9)$$

where $|\boldsymbol{k}| = k_R$ and $\theta_{\boldsymbol{k}} \equiv \arctan(k_y/k_x)$. The pseudospins of these states form a rotating-in-time "vortex" in the $x$-$y$ plane, see Fig. 1b. In the low-doping limit ($\Delta n \to 0$) and in the regime where cooling dominates over heating processes ($\kappa \ll \Delta n^2$), the upper Floquet band has a significant population only near the band's bottom. Above a critical interaction strength, we expect the self-consistent solution to converge to a GFN steady-state where the electrons localize around a single spontaneously chosen momentum on the ring (Fig. 1c). Subsequently, due to the time-dependent pseudospin-momentum locking in Eq. (9), the pseudospins of the electrons will be synchronized. This implies that the "in-plane" ($x$-$y$) components of $\boldsymbol{h}(t)$ should take the form of a rotating (circularly polarized) field, with its dominant harmonic given by $\boldsymbol{h}_1^{(xy)} \approx h_1(\hat{\boldsymbol{x}} - i\hat{\boldsymbol{y}})/\sqrt{2}$. In our analysis we use $|\boldsymbol{h}_1^{(xy)}|$ as the diagnostic for spontaneous symmetry breaking.

We note that in both the symmetry broken and un-broken phases, the system exhibits an oscillating $z$-component of the magnetization $\boldsymbol{h}(t)$. The $z$-component of the harmonic $\boldsymbol{h}_1$ renormalizes the amplitude and phase of the drive [see text below Eq. (1)]. As we will show below, throughout the parameter regime of interest this renormalization remains weak. Therefore, in estimating the critical interaction strength below, we neglect this component and keep only $\boldsymbol{h}_1^{(xy)}$.

We now seek the minimal interaction strength, $U_c$, required to achieve spontaneous symmetry breaking for finite values of $\kappa$ and $\Delta n$. To make progress, we approximate the distribution of electrons in the upper Floquet band by a Fermi-Dirac distribution with an effective chemical potential, $\mu_e$, measured from the bottom of the upper band, and temperature (measured in energy units), $T_e$. Analogously, we parametrize the hole distribution in the lower Floquet band by an effective chemical potential $\mu_h$, measured from the top of the lower band, and temperature $T_h$. Such a fit well-approximates the distributions in the limit of low density (see ref. [91] and Supplementary Fig. 9). Note that the electron and hole populations are generically described by finite effective temperatures, even when the baths are at zero temperature.

In Eq. (7) above, we found the total densities of electrons and holes in the upper and lower Floquet bands, $n_e$ and $n_h$, respectively. However, a given pair of values for $n_e$ and $n_h$ can be obtained for a continuous family of choices of $\mu_{e/h}$ and $T_{e/h}$. Below we first derive a general result for the critical interaction

strength $U_c$, parametrized by the chemical potentials and temperatures that are realized. Later, we will discuss how to determine the values of $\mu_{e/h}$ and $T_{e/h}$ in the steady-state.

To find $U_c$, assuming the transition is continuous, we solve Eq. (2) by expanding the expectation value on its RHS to linear order in the amplitude of the in-plane ($x$-$y$) component of the magnetization, $|\boldsymbol{h}_1^{(xy)}|$, which we take to be circularly polarized. Note that the RHS of Eq. (2) depends on $\boldsymbol{h}(t)$ through the steady-state distribution, $f_{\boldsymbol{k}\nu}$, defined in the basis of the eigenstates of $H_{MF}$ [which also depend on $\boldsymbol{h}(t)$], see Eq. (3). Given that the effective temperature and chemical potential weakly depend on $\boldsymbol{h}_1^{(xy)}$, the dominant dependence of $f_{\boldsymbol{k}\nu}$ on $\boldsymbol{h}_1^{(xy)}$ arises from the eigenstates and eigenvalues of $H_{MF}$.

Expanding the RHS of Eq. (2) to linear order in $h_1$ yields three terms: (i) a contribution corresponding to a full lower Floquet band, and the contributions of (ii) the electrons and (iii) the holes in the upper and lower Floquet bands, respectively. We use the assumed Fermi-Dirac distribution functions for electrons and holes to evaluate each of the terms analytically (for the full derivation see Supplementary Note 2), yielding an expression for the critical interaction strength:

$$\tilde{U}_c^{-1} = \tilde{U}_{fb}^{-1} + \tilde{U}_{ex}^{-1}\left(\frac{\tilde{\Theta}(\mu_e/T_e)}{\tilde{n}_e} + \frac{\tilde{\Theta}(\mu_h/T_h)}{\tilde{n}_h}\right). \qquad (10)$$

Here $\tilde{U}_c = \mathcal{A}_R U_c/\delta E$, $\tilde{n}_e = n_e/\mathcal{A}_R$, and $\tilde{n}_h = n_h/\mathcal{A}_R$ are the normalized interaction strength and population densities, respectively. (Recall that $\mathcal{A}_R$ is the area in reciprocal space enclosed by the resonance ring and $\delta E = \hbar\Omega - E_g$.) The dimensionless function $\tilde{\Theta}$ will be defined below. The contribution to the inverse of $\tilde{U}_c$ of type (i) above is given by $\tilde{U}_{fb}^{-1}$. For a hypothetical state with a full lower Floquet band and an empty upper Floquet band, the critical interaction strength would be equal to $\tilde{U}_{fb}$. The contributions to $\tilde{U}_c^{-1}$ of types (ii) and (iii) are captured by the terms proportional to $\tilde{U}_{ex}^{-1}$ in Eq. (10). At finite doping, and/or with a finite density of electron–hole excitations, these terms reduce the critical interaction strength. In the derivation of Eq. (10) we obtain explicit expressions for these coefficients, $\tilde{U}_{ex} = 4\pi^4\delta E/\Delta_F$ and $\tilde{U}_{fb} = 2\pi^2\left[\log\left(\frac{8E_{BW}\delta E}{\Delta_F^2}\right) - 1\right]^{-1}$, where $E_{BW}$ is a high-energy cutoff representing the bandwidth of the semiconductor, see Supplementary Note 2.

The enhancement of the density of states at the ring extrema of the Floquet bandstructure affects $\tilde{U}_c$ through the terms of type (ii) and (iii) in Eq. (10). The unitless function $\tilde{\Theta}(x)$ that appears in this term has the form of a "smeared" step function that drops to zero when its argument is negative, and saturates to 1 in the opposite limit, with a smooth cross-over whose width is $\mathcal{O}(1)$. Therefore, the contribution of type (ii) is governed by a competition between two effects: on the one hand, for this term to be significant, a small density of electrons is required. On the other hand, to achieve $\tilde{\Theta}(\mu_e/T_e) \approx 1$ the distribution of the electrons in the upper Floquet band is required to have a sharp Fermi surface (which is realized for $\mu_e/T_e \gg 1$). When these conditions are met, the critical interaction strength is suppressed due to the divergence of the DOS at the ring minimum. Similar considerations hold for the contribution of type (iii) arising from holes in the lower Floquet band.

Equation (10) is a non-equilibrium analogue of the Stoner criterion[97,98], which gives the critical interaction strength for spontaneous symmetry breaking in the steady-state of the system. The criterion crucially depends on the effective chemical potentials and temperatures of electrons and holes in the steady state, which are controlled by the interactions both within the

system and between the system and its environment. As discussed above, when the electrons in the upper band form a low-density population with a sharp Fermi surface (such that $\tilde{\Theta} \approx 1$), the critical interaction strength $U_c$ may be reduced. Such a suppression of $U_c$ is particularly important for ensuring the possibility that a low-temperature symmetry-broken steady-state can arise in the non-equilibrium system, as the heating rate due to electron–electron scattering scales as $U^2$ [see Eq. (6)]. In the next section, we will analyze the phase diagram of the system using both numerical simulations and further analysis based on the rate equation approach.

**Phase diagram and numerical simulations**. In this section, we introduce a lattice model whose effective description for momenta near the $\Gamma$-point is given by Eq. (1). Our motivation is to demonstrate symmetry breaking from a full self-consistent solution of the coupled kinetic and Floquet mean-field equations in Eqs. (2)–(4). In addition, we seek to validate the suppression of $U_c$ due to the enhanced density of states near the resonance ring [exhibited by the term proprotional to $\tilde{U}_{ex}^{-1}$ in Eq. (10)]. To this end, we extend the Hamiltonian in Eq. (1) to the entire Brillouin zone of a square lattice with primitive lattice vectors $\boldsymbol{a}_1 = (a, 0)$ and $\boldsymbol{a}_2 = (0, a)$. We consider nearest and next nearest neighbor hopping, described by the modified Hamiltonian $H_0(\boldsymbol{k}) = \boldsymbol{d}(\boldsymbol{k})\cdot\boldsymbol{\sigma}$, where $\boldsymbol{d} = (d_x, d_y, d_z)$, with $d_{x(y)}(\boldsymbol{k}) = A\sin(ak_{x(y)}) + A'[\sin(\boldsymbol{r}_1\cdot\boldsymbol{k}) \pm \sin(\boldsymbol{r}_2\cdot\boldsymbol{k})]$, $d_z(\boldsymbol{k}) = E_g/2 - B[\cos(ak_x) + \cos(ak_y) - 2]$, and Hubbard interaction $\mathcal{V}_q = U/\varpi$. The coefficients $A'$ and $B'$ denote the next-nearest neighbor hopping along the vectors $\boldsymbol{r}_{1,2} = a(\hat{\boldsymbol{x}} \pm \hat{\boldsymbol{y}})$. In the numerical simulations we set $A' = A/4$ and $B' = B/4$. Such a fine-tuning of the next-nearest neighbor hopping parameters helps to minimize the terms that break the U(1) symmetry of the resonance ring to $C_4$ due to lattice effects. These effects lift the degeneracy at the resonance ring, thus cutting off the divergence of the DOS. Note that the form of $H_0(\boldsymbol{k})$ used for the numerical simulation agrees with the Hamiltonian $H_0(\boldsymbol{k})$ in Eq. (1) for momenta near the $\Gamma$-point, with $A = 2\lambda_0/3a$, and $B = 2\hbar^2/3m_*a^2$. We consider the case $E_g, B > 0$ (such a choice provides non-inverted bands of the semiconductor), and restrict $\hbar\Omega > E_g/2 + 4B$ to ensure that $2\hbar\Omega$ is larger than the total bandwidth, such that there are no second and higher order resonances in the numerical simulation.

Using the lattice model within the mean-field approximation, we numerically solved the rate and the mean-field equations for the steady state in a self-consistent manner according to the procedure described between Eqs. (2)-(4). To this end, we computed the occupation function $f_{\boldsymbol{k}\nu}$ and the scattering rates $I_{\boldsymbol{k}\nu}^s$, $I_{\boldsymbol{k}\nu}^\ell$, and $I_{\boldsymbol{k}\nu}^{ee}$ using a non-uniform grid of 8008 points in momentum space, with enhanced resolution in the vicinity of the resonance ring. We evaluated the scattering rates using Fermi's golden rule with the electron–phonon coupling matrix $\mathcal{M}_s(\boldsymbol{q}_\parallel, \omega) = g_s|\boldsymbol{q}_\parallel|/\sqrt{\omega}$, and electron-photon coupling matrices for two orthogonal photon polarizations, $\mathcal{M}_\ell^{(1)} = g_\ell\sigma^x$ and $\mathcal{M}_\ell^{(2)} = g_\ell\sigma^y$. The DOSs for spontaneous emission of acoustic phonons ($p = s$) and photons ($p = \ell$), traced over the out-of-plane momentum, are given by $\rho_p(\omega, \boldsymbol{q}_\parallel) = \rho_p^0\,\omega/\sqrt{\omega^2 - |v_p\boldsymbol{q}_\parallel|^2}$ when $\omega > v_p|\boldsymbol{q}_\parallel|$ and $\rho_p = 0$ otherwise, where $\omega$ is the frequency of the emitted phonon or photon and $\boldsymbol{q}_\parallel$ is the in-plane component of its momentum. The constants $g_{s(\ell)}$ and $\rho_{s(\ell)}^0$ are material-dependent parameters. In the simulations we tune $g_{s(\ell)}$ and $\rho_{s(\ell)}^0$ to explore their roles in determining the steady states, and to effectively tune the heating parameter $\kappa$ for comparison with our analytical results. In the numerical results presented in the main text, we focus on the regime $\kappa_{ee} \ll \kappa_{ph}$, where Floquet-Umklapp electron–electron scattering processes do not significantly

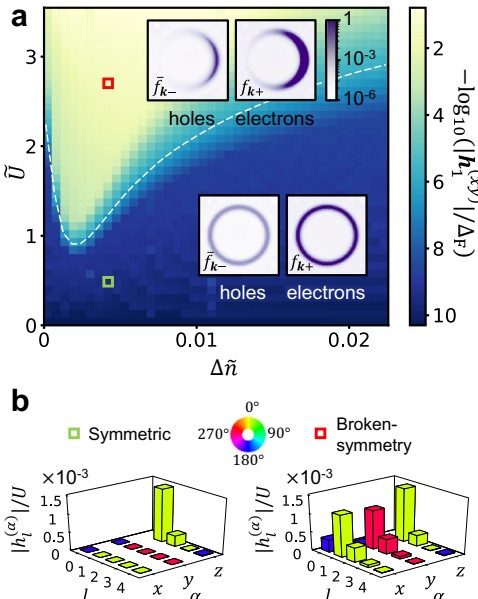

**Fig. 2 Floquet phase diagram. a** Spontaneous magnetization strength, $|\boldsymbol{h}_1^{(xy)}|$, obtained from the self-consistent mean-field calculation, as a function of a normalized electron doping, $\Delta\tilde{n} \equiv \Delta n/\mathcal{A}_\mathrm{R}$ and normalized interaction strength, $\tilde{U} \equiv \mathcal{A}_\mathrm{R}U/\delta E$. The dashed white line represents the phase boundary, corresponding to the critical interaction strength $\tilde{U}_\mathrm{c}$, calculated from Eq. (10). The insets show the electron and hole steady-state distributions (respectively $f_{\boldsymbol{k}+}$ and $\bar{f}_{\boldsymbol{k}-} \equiv 1 - f_{\boldsymbol{k}-}$) in the momentum domain near the resonance ring, for $\Delta\tilde{n} = 0.004$, $\tilde{U} = 0.44$, indicated by a green square, in the symmetric phase, and $\Delta\tilde{n} = 0.004$, $\tilde{U} = 2.66$, indicated by a red square, in the symmetry-broken phase. **b** Harmonics of the self-consistent magnetization $\boldsymbol{h}(t) = \mathrm{Re}\left[\sum_{l,\alpha}\hat{\boldsymbol{e}}_\alpha h_l^{(\alpha)} e^{il\Omega t}\right]$, where $\hat{\boldsymbol{e}}_\alpha = \hat{\boldsymbol{x}}, \hat{\boldsymbol{y}}, \hat{\boldsymbol{z}}$. We plot $|h_l^{(\alpha)}|/U$ corresponding to the first five harmonics ($l = 0, 1, 2, 3, 4$) at the two points on the phase diagram indicated by the red and green squares in panel **a**. The heights and the colors of the bars respectively indicate the amplitudes and phases of the harmonics. The color scale for the phase is shown at the top of the panel. Note that we omit $|h_0^{(z)}|$, which is responsible for the bandgap renormalization of the system in the absence of the drive, see main text.

contribute to the heating rate. We obtain qualitatively similar results in the regime of $\kappa_\mathrm{ee} \gtrsim \kappa_\mathrm{ph}$, see Supplementary Note 6.

In each iteration of the algorithm, we numerically compute the magnetization $\boldsymbol{h}(t)$ via Eq. (2). To improve the precision of the momentum integral, we first fit the electron and hole distributions to Fermi functions, then perform the integration using the fits interpolated to a finer grid. In the simulations, we allow for the magnetization to develop components up to the fifth harmonic of the driving frequency. As discussed below Eq. (2), we discard the constant in-time component in the $z$ direction, which simply renormalizes the parameters of the underlying band structure.

In Fig. 2a we show the non-equilibrium phase diagram of the system in the plane of doping, $\Delta n$, and interaction strength, $U$. The bath parameters are fixed with values that yield $\kappa_0 a^4 \approx 10^{-9}$, where the bare heating parameter, $\kappa_0$, denotes the value of the heating parameter $\kappa$ at $U = 0$ and half-filling (see Supplementary Note 5a for details). The color scale in Fig. 2a indicates the magnitude of spontaneous magnetization, $|\boldsymbol{h}_1^{(xy)}|$, for a lightly electron-doped system. The figure shows two distinct phases: a symmetric phase (blue), $|\boldsymbol{h}_1^{(xy)}| = 0$, and a broken-symmetry phase (yellow), $|\boldsymbol{h}_1^{(xy)}| > 0$.

We present characteristic particle distributions well-inside of each phase in the insets to Fig. 2a. In the paramagnetic (symmetry-preserving) phase, the electron and hole populations exhibit uniform occupation of states around the resonance ring. In the GFN (broken symmetry) phase, the electron and hole populations are concentrated on one side of the resonance ring. The magnitudes of the harmonics of $\boldsymbol{h}(t)$ for the same representative states in the two phases are shown in Fig. 2b. Here it is evident that in the broken symmetry phase the first harmonic $\boldsymbol{h}_1$ gives the dominant contribution, yet the DC component and second harmonic are substantial. Although present, as discussed below Eq. (8), these harmonics do not significantly affect the Floquet mean-field band structure.

The boundary between the phases occurs at a critical interaction strength $U_\mathrm{c}$. The dependence of $U_\mathrm{c}$ on the doping $\Delta n$ can be explained using Eq. (10). However, to use Eq. (10) we first need to know how $\mu_{e(h)}/T_{e(h)}$ and $n_{e(h)}$ depend on $\Delta n$ and other parameters of the model. The electron and hole densities $n_{e(h)}$ found from the phenomenological rate equation treatment are given in Eq. (7). We now seek two additional equations to fix the ratios $\mu_e/T_e$ and $\mu_h/T_h$ for these electron and hole populations, respectively. (Recall that the same values of $n_e$ and $n_h$ can be obtained from a continuous family of values of $\mu_e/T_e$ and $\mu_h/T_h$.) Note that $\mu_{e(h)}/T_{e(h)}$ and $n_{e(h)}$ depend on $U$ through their dependence on $\kappa$. Therefore, the RHS of Eq. (10) implicitly depends on $U_\mathrm{c}$. For simplicity, as in the numerical simulations that lead to Fig. 2, here we focus on the case of $\kappa_\mathrm{ee} \ll \kappa_\mathrm{ph}$, where the heating parameter $\kappa$ can be treated as a $U$ independent parameter (see text below Eq. (6) for definitions).

Connecting back to $U_\mathrm{c}$ given by Eq. (10), recall that $\tilde{\Theta}(\mu_e/T_e) = \mathcal{O}(1)$ when $\mu_e/T_e > 1$, leading to a suppression of $U_\mathrm{c}$. In this situation, the population of electrons in the upper band exhibits a sharp Fermi surface. We refer to such a state as a degenerate electronic Floquet metal (EFM). Alternatively, if $\mu_e/T_e < 0$, the effective chemical potential lies in the Floquet gap and the electronic distribution corresponds to a non-degenerate Fermi gas. We refer to such a state as an electronic Floquet insulator (EFI). In this state, $\tilde{\Theta}(\mu_e/T_e)$ is small.

We now discuss the factors that determine the value of $\mu_e/T_e$ and which phase (EFM or EFI) is achieved in the steady state. The EFM phase is established when the intraband cooling of excited electrons is more efficient than the relaxation of electrons from the upper to the lower Floquet band. In this case, electrons excited from the lower to the upper Floquet band via Floquet-Umklapp processes quickly relax to the bottom of the upper Floquet band, where a Fermi sea is formed. The flow of electrons into the Fermi sea in the upper Floquet band is balanced by phonon-assisted annihilation of electrons in the Fermi sea with holes in the lower Floquet band. The balance between these interband and intraband rates can be analyzed by extending the rate equation treatment, expressed in Eq. (6), to include an energy-resolved treatment of the electron and hole populations, see ref. [91] and Supplementary Note 3b.

Deep in the EFM phase, and for $U \lesssim U_\mathrm{c}$, the extended rate equation treatment yields $\mu_e/T_e \approx x_e^{1/4}$, where $x_e \equiv \zeta n_e^6/(v_s^3 \kappa)$, and we estimate $\zeta \approx C\hbar^5/(\Delta_\mathrm{F} m_*^4 k_\mathrm{R}^3)$, where $C$ is a constant of $\mathcal{O}(10^{-3})$ (see Supplementary Notes 3h,3i for the full details). Since the EFM phase corresponds to large $\mu_e/T_e$ and therefore large $x_e$, this phase is favored at large electron density (large doping), small sound velocity $v_s$, and low values of the heating parameter $\kappa$. In particular, lower sound velocities facilitate intraband cooling, as this leads to an enhancement of the density of states for low-frequency phonons.

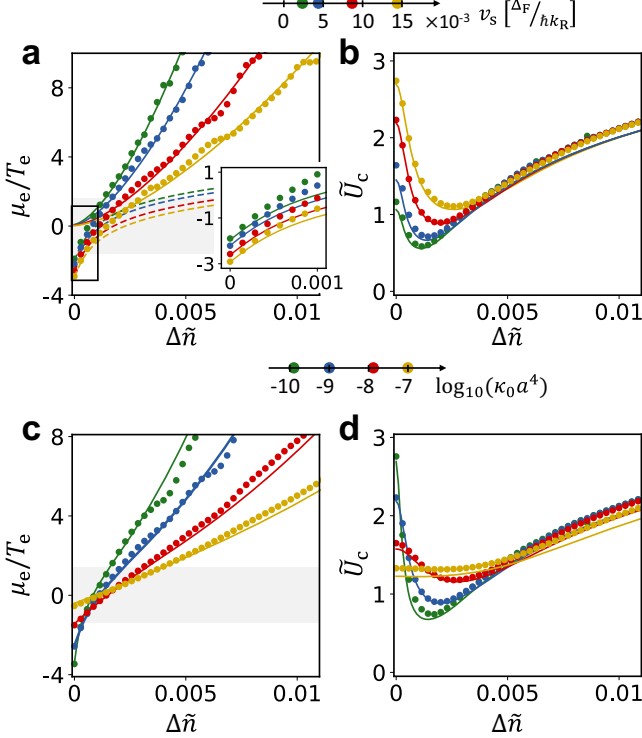

**Fig. 3 Phase boundary and crossover from EFM to EFI regimes. a**
Numerically obtained ratio of effective chemical potential to temperature of
the electronic population in the upper Floquet band, as a function of the
normalized doping (data points). The data are extracted from the steady-
state solution to Eq. (4) for $\kappa_0 a^4 \approx 10^{-9}$ and four values of speed of sound,
$v_s$. Solid and dashed lines represent the results of the extended rate
equation treatment for the EFM and the EFI phases, respectively. The full
set of curves is generated using the same value of the single fit parameter $\zeta$
(see text for definition). The shaded area indicates the EFI-to-EFM
crossover range where the $\tilde{\Theta}$-function in Eq. (10) rises from 0 to 1. Inset:
Zoom-in on the low-doping regime (enclosed by a black frame in the main
panel). Solid lines correspond to the analytical curves for the EFI phase.
**b** Critical interaction strength $U_c$ extracted for the same data set as in panel
**a** (data points). Solid lines represent $U_c$ calculated from Eq. (10), where for
the values of $\mu_e/T_e$ we used a function that interpolates between the
analytical results deep in the EFM and EFI phases. We use the same value
of $\zeta$ as in panel **a**, and two additional fitting parameters $\tilde{U}_{ex}$ and $\tilde{U}_{fb}$.
**c, d** Results for $\mu_e/T_e$ and $U_c$, extracted in the same manner as in panels
**a** and **b** (data points), for $v_s = 0.0086 \, \Delta_F/\hbar k_R$, and four values of $\kappa_0$. Solid
lines in the two panels show the interpolated values of $\mu_e/T_e$ and the
resulting $U_c$. All fitting parameters are the same as in panels **a** and **b**.

The EFI phase is obtained in the opposite limit, where interband
relaxation is more efficient than intraband cooling. In this case, the
extended rate equation treatment yields $e^{\mu_e/T_e} \approx x_e^{1/5}$. The extended
rate equation treatment can also be used to characterize the hole
population in the lower Floquet band. We find that for electron
doping, $\Delta n > 0$, the holes form a non-degenerate Fermi gas for all
parameter values within our model. Fast, quasienergy-conserving
electron–hole scattering processes tend to equalize the electron and
hole temperatures, $T_e$ and $T_h$.

Figure 3 shows a comparison of $\mu_e/T_e$ and $U_c$ extracted from
numerical simulations (data points), to the analytical estimates
obtained from the extended rate equations discussed above. We
obtain $U_c$ using a numerical analogue of the procedure leading to

Eq. (10). Specifically, we compute the expectation value on the
RHS of Eq. (2), using the steady state obtained from Eq. (4) for a
system whose electronic Hamiltonian corresponds to $\hat{\mathcal{H}}_{MF}(t)$ in
Eq. (3). In this procedure for obtaining $U_c$, we use a prescribed
form of $\boldsymbol{h}(t)$ with a single non vanishing harmonic $\boldsymbol{h}_1^{(xy)}$ of small
magnitude in $\hat{\mathcal{H}}_{MF}(t)$, see Eq. (8). The dashed white line in Fig. 2a
shows $U_c$ extracted using the above procedure on top of the phase
diagram obtained from the full self-consistent numerical simula-
tions for the same parameters.

In Fig. 3a we show $\mu_e/T_e$ as a function of doping $\Delta n$ for several
values of phonon sound velocity $v_s$. We extract $\mu_e$ and $T_e$ from the
numerical simulations described in the previous paragraph by fitting
the electron steady-state distribution in the upper Floquet band to a
Fermi function with respect to the quasienergies of the mean-field
Hamiltonian [Eq. (3)]. The fit lines correspond to the analytical
forms for $\mu_e/T_e$ obtained from the extended rate equation treatment.
The solid and dashed lines correspond to the forms for $\mu_e/T_e$ in the
EFM and EFI regimes, respectively. The only freedom in these fits is
the parameter $\zeta$ in the definition of $x_e$, which was given the same
value across all of the curves shown. The extracted value of $\zeta$ is of the
same order of magnitude as the analytical estimate given above.

In Fig. 3c we again show $\mu_e/T_e$ as a function of $\Delta n$, this time
highlighting the dependence on the value of the bare heating
parameter $\kappa_0$. In this plot, the fit lines are given by a function that
interpolates between the analytical results for the asymptotic
behavior in the EFM and EFI regimes: $\mathcal{F}_j[\lambda(\mu_e/T_e)] = x_e^{\eta}$, where
$\mathcal{F}_j$ is the complete Fermi-Dirac integral, $\eta = 0.174$, $j = -0.3$, and
$\lambda = 0.871$. These parameter values are fixed by demanding that
$\mu_e/T_e$ displays the correct dependence on $x_e$ deep in the EFI
($e^{\mu_e/T_e} \sim x_e^{1/5}$) and EFM ($\mu_e/T_e \sim x_e^{1/4}$) phases. The value of $\zeta$
used in $x_e$ is the same as in Fig. 3a.

In Fig. 3b, d we show $U_c$ as a function of $\Delta n$ for different values
of $v_s$ and $\kappa_0$. The data points are obtained from the numerical
procedure discussed above. To obtain the fit lines, we use the
interpolated values of $\mu_e/T_e$ in Eq. (10). We additionally use $\tilde{U}_{ex}$
and $\tilde{U}_{fb}$ as fitting parameters. The same values of these
parameters were used in all curves shown. The values used for
the fits are close to those obtained from the formulas given below
Eq. (10). For the contribution of the holes, we used the same
interpolating function, with $x_h$ replacing $x_e$. Here $x_h$ is defined in
the same manner as $x_e$, but with $n_h$ replacing $n_e$ (with the same
value of the parameter $\zeta$). Note that holes are in the analogue of
the EFI phase for any $\Delta n > 0$, and hence the value of $\tilde{\Theta}(\mu_h/T_h)$ is
small throughout the regime studied.

As is evident in the phase diagram in Fig. 2a, $U_c$ obtains a minimal
value at an optimal value of the doping, which we denote $\Delta n_*$. Using
the extended rate equation treatment, we estimate
$\Delta n_* = C_*(v_s^3 \kappa/\zeta)^{1/6}$, where $C_*$ is a constant of $\mathcal{O}(1)$, see
Supplementary Note 3j. The corresponding minimal interaction
strength is given by $\tilde{U}_c^{min} \approx \tilde{U}_{ex} \Delta n_* / \mathcal{A}_R$ for $\kappa \ll k_R^4$. For $\Delta n > \Delta n_*$,
electrons in the upper band are in the EFM phase, and exhibit a
sharp Fermi surface (note the corresponding values of $\mu_e/T_e$ in
Fig. 3a). As explained below Eq. (10), the existence of a Fermi surface
tends to reduce the critical interaction strength. However, as the
doping increases, the phase transition requires stronger interactions
as the density of states at the Fermi surface decreases. Below the
optimal doping, $\Delta n < \Delta n_*$, the electrons in the upper Floquet band
are in the EFI regime, which has no Fermi surface. Thus, the
suppression of $U_c$ is lost for $\Delta n < \Delta n_*$.

**Discussion**
In this work, we demonstrated a mechanism for realizing elec-
tronic liquids crystals in two-dimensional electronic systems

through time-periodic driving. The phase that we find exhibits GFN order associated with the spontaneous breaking of U(1) symmetry in both pseudospin and orbital degrees of freedom. Above the critical interaction strength, the Fermi sea becomes highly anisotropic and occupies a limited sector of the ring minimum of the Floquet bands, see Figs. 1 and 2. Due to pseudospin-momentum locking, the Fermi sea in the symmetry broken phase exhibits a finite magnetization that rotates with the frequency of the drive [cf. Eq. (8)]. We note that the system Hamiltonian is only nearly U(1) symmetric as it is defined on a lattice with $C_4$ symmetry. We verified that the terms that reduce the symmetry to $C_4$ do not affect the mean-field phase diagram by comparing the lattice model to the fully U(1)-symmetric analytical model. Even when these terms are weak, at energies of interest they serve as a relevant perturbation that will pin the location of the Fermi pocket on the resonance ring, and suppress long-wavelength fluctuations of the order.

Our analysis has been carried out on a model system with a two-component pseudospin degree of freedom, whose band-structure is described by $H_0(\boldsymbol{k})$, see Eq. (1). For $E_g > 0$, the model $H_0(\boldsymbol{k})$ lacks time-reversal symmetry. The model can also be taken to describe half of the degrees of freedom of a time-reversal invariant system. Our analysis can be straightforwardly extended to include the relevant time-reversal partner degrees of freedom. In this situation, there are many more possibilities for how the system may order. As one example, in Supplementary Note 4 we describe a mean-field treatment that shows an instability towards an order in which the magnetizations of the two time-reversal partners are aligned, which yields a breaking of the time-reversal symmetry. We leave a more elaborate study of this interplay, as well as other possible phases[39,41], for future investigation.

For simplicity, throughout the paper, we considered a driving whose form is described below Eq. (1). It is interesting to consider driving with circularly polarized light, which, like the drive we studied, preserves the U(1) symmetry of the system and uniformly opens a gap all the way around the resonance ring. Depending on the handedness of the drive, the pseudospin may wind twice or zero times around the resonance ring[67]. In the case of double winding, each direction of the pseudospin in the x-y plane corresponds to two momentum points on the resonance ring. Therefore, in this case, an analogous GFN phase would exhibit two electron pockets occupying opposite sectors of the ring minimum.

Here, we considered a model with short-ranged interactions. Long-ranged interactions such as Coulomb or dipole interactions may lead to additional possibilities for the steady state, such as phases with broken translation symmetries similar to Wigner crystals[12] or quasicrystals[99]. It would be interesting to investigate the possibility that the order parameter may exhibit a more exotic temporal structure, such as oscillations with a frequency that is different than the frequency of the drive[100,101].

We now put our results in an experimentally relevant context. To this end, in the following we will make several estimates, using model parameters which are given in Supplementary Note 7. We first estimate the value of the heating parameter $\kappa$, employing the definition of $\kappa$ appearing below Eq. (6). We base our estimate on typical scattering rates measured in semiconductors[102]. We start with the phonon-assisted interband scattering rate, which tends to reduce the number of excited electrons in the upper Floquet band. This is estimated in Supplementary Eq. (65) as $\Lambda_{\text{inter}} \approx \hbar/(\tau_\Lambda^2 \Delta_F \mathcal{A}_{\text{BZ}})$. Here $\mathcal{A}_{\text{BZ}}$ is the reciprocal-space area of the Brillouin zone and $\tau_\Lambda^2 = \hbar^2/(12\pi^3 g_s^2 \rho_s^0 k_R^2)$. The primary source of heating is radiative recombination, which predominantly occurs between states with inverted band indices, i.e., inside the resonance ring. We estimate this rate by $\Gamma_\ell \approx \mathcal{A}_R/\tau_\ell$, where $\mathcal{A}_R$ is the area inside the resonance ring and $\tau_\ell$ is a typical

time for the radiative recombination which is on the order of 1 ns. In addition, electron–phonon scattering also contributes to heating through Floquet-Umklapp scattering processes. However, in the Supplementary Note 3d we show that these processes lead to a negligible contribution to the dynamics. Likewise, for small $V/(\hbar\Omega)$ electron–electron interactions also give a sub-dominant contribution to the heating dynamics (see below). Assuming $V \sim 10$ meV, and $\mathcal{A}_R/\mathcal{A}_{\text{BZ}} \approx 4 \times 10^{-3}$, and for typical semiconductor phonon parameters (see Supplementary Note 7), we estimate $\kappa \approx \Gamma_\ell/\Lambda_{\text{inter}} \approx 3 \times 10^{-7} \mathcal{A}_{\text{BZ}}^2$.

As stated above, the electron–hole pair generation (heating) rate $\Gamma_{ee}$ due to photon-assisted electron–electron scattering is small when $V/(\hbar\Omega)$ is small. To lowest order in $(V/\hbar\Omega)^2$, these processes predominantly excite a pair of electrons from the lower to the upper Floquet band, accompanied by the absorption of one photon from the driving field. Due to the pseudospin structure of the Floquet states the dominant scattering processes involve one electron that is scattered from the interior of the resonance ring to the exterior, and vice versa for the other electron. Therefore, we expect the scattering rate to be proportional to the squared area of the resonance ring. After averaging over initial and final momenta, we estimate $\Gamma_{ee}(U) = \frac{\mathcal{A}_R^2 m_* U^2}{2\pi^4 \hbar^3}\left(\frac{V}{\hbar\Omega}\right)^2$. Here $m_*$ is the effective mass of the semiconductor [Eq. (1)].

Given that the heating rate depends on the interaction strength, it is important to check how the critical interaction strength is affected by the presence of Floquet-Umklapp scattering due to electron–electron interactions. The critical interaction strength is not significantly changed when $\kappa_{ee}(U_{\text{fb}}) \ll \kappa_{\text{ph}}$, where $\kappa_{ee}(U) = \Gamma_{ee}(U)/\Lambda_{\text{inter}}$ [see Eq. (10) and Supplementary Note 6 for supporting numerical simulations] and $\kappa_{\text{ph}}$ is the value of $\kappa$ for $\Gamma_{ee} = 0$. The estimate for $\Gamma_{ee}$, when $\mathcal{A}_R$ is small, shows that for realistic parameter choices, the above conditions can be indeed satisfied. We note that short-range interactions corresponding to this regime can be obtained using screening gates placed near the 2D electronic system.

We next estimate the total energy flux density transferred to the phonon and photon heat baths. We approximate this energy flux density by $W = \Gamma_\ell \hbar\Omega$. In turn, the energy dissipated to photons and phonons is given by $W_\ell = \Gamma_\ell E_g$ and $W_s = \Gamma_\ell(\hbar\Omega - E_g)$, respectively, such that $W = W_\ell + W_s$. (Here, $E_g$ is the band gap of the semiconductor.) Using the same parameters as above and assuming a sample of area $L^2$, where $L = 5\,\mu$m, we arrive at $W_\ell L^2 \approx 4$ mW and $W_s L^2 \approx 60\,\mu$W. The power emitted to phonons can be dissipated by standard cryogenic refrigerators operating at temperatures of a few degrees of Kelvin.

Finally, we estimate the upper and lower bounds for the drive intensity for realizing the GFN phase. The lower bound is found as the value at which the Floquet gap equals the scattering rate, $\Delta_{\min} = \hbar/\tau_{\text{scat}}$. Below this value, the diagonal ensemble of Floquet states is not a proper fixed point for the steady state distribution[82]. In the regime discussed above, for which the heating rate is dominated by radiative recombination, we estimate $\tau_{\text{scat}}$ using Eq. (6). This yields $\tau_{\text{scat}}^{-1} \approx \Gamma_\ell/n_e$, see Supplementary Note 7. We obtain $n_e$ from from Eq. (7). For a system at half-filling, this gives $\tau_{\text{scat}}^{-1} = \sqrt{\Gamma_\ell \Lambda_{\text{inter}}}$. The Floquet gap then equals the scattering rate for $\Delta_{\min} \approx 0.2$ meV.

We estimate the upper bound on the Floquet gap as the value at which heating due to electron–electron interactions becomes dominant over that resulting from radiative recombination, i.e., beyond $\Gamma_{ee} = \Gamma_\ell$. Beyond this point, the heating rate grows rapidly with the driving amplitude. For $\tilde{U} \approx 3$, we arrive at $\Delta_{\max} \approx 0.2$ eV. The corresponding applied field strength and intensity for circularly polarized light[67] are given by $\mathcal{E} \approx 4 \times 10^7$ V/m and $\mathcal{I} \approx 2 \times 10^{12}$ W/m$^2$, respectively, compatible with recent experiments on Floquet engineered materials[63,69]. Note that compared to these experiments, in the setup that we propose heating is reduced by

working in a more favorable frequency regime. Specifically, we consider a frequency comparable to the band gap of a wide-gap semiconductor (rather than working at much lower mid-infrared frequencies). For a fixed value of the Floquet gap, Floquet-Umklapp scattering due to electron–electron and electron–phonon interactions is suppressed at large frequencies.

The phenomenon we discussed can be realized in 2D Dirac systems such as transition metal dichalcogenides and semiconductor quantum wells. To ensure that Floquet-Umklapp processes are suppressed, it is beneficial to use large bandgap materials. In gapless Dirac system such as graphene, driving may induce similar ring-like Floquet-band extrema[73]. We leave the exploration of particle dynamics and symmetry breaking in such systems to future studies. The use of periodic driving to create ring extrema in Floquet bands may be utilized to study exotic phases of fermions and bosons in cold atom systems[23,45,66]. In particular, it would be interesting to investigate the possibility to use buffer gases in cold atom systems to serve as the heat baths needed for stabilizing the broken-symmetry phases discussed in this work.

## Data availability
The data that support the findings of this study are available from the corresponding author upon reasonable request.

## Code availability
The code that supports the findings of this study is available from the corresponding author upon reasonable request.

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

## Acknowledgements

We would like to thank Ehud Altman, Vladimir Kalnizky, Gil Refael, and Ari Turner for illuminating discussions and David Cohen and Yan Katz for technical support. N.L. acknowledges support from the European Research Council (ERC) under the European Union Horizon 2020 Research and Innovation Programme (Grant Agreement No. 639172), and from the Israeli Center of Research Excellence (I-CORE) "Circle of Light". M.R. gratefully acknowledges the support of the European Research Council (ERC) under the European Union Horizon 2020 Research and Innovation Programme (Grant Agreement No. 678862) and the Villum Foundation. M.R. and E.B. acknowledge support from CRC 183 of the Deutsche Forschungsgemeinschaft. G.K.G. acknowledges support from Israel Council for Higher Education.

## Author contributions

I.E. and G.K.G. performed the numerical and analytical calculations. I.E., G.K.G., E.B., M.R. and N.L. analyzed the data and wrote the manuscript.

## Competing interests

The authors declare no competing interests.
