## [Peer Review File · Nature Communications]

REVIEWER COMMENTS

Reviewer #1 (Remarks to the Author):

In the present manuscript entitled "Electronic Liquid Gyro-Crystals," authors considered interesting phases of interacting fermions, which appear in systems with moat spectra. This type of spectra appears in two-dimensional materials with strong

Rashba-type spin-orbit coupling. In this type of system, as shown in the references [38,39] of the paper, the ground state is sensitive to states' density, leading to novel phases. The crucial point here is the observation that the ring-minimum in such systems leads to a large degeneracy and a divergent DOS at energies approaching the band's bottom. The competition of inter-particle interactions with kinetic energies at low densities may lead to symmetry-broken phases. These phases exhibit spontaneously broken rotational symmetry, anisotropic Fermi surfaces. All these properties are already well known,

but in this paper, authors present an exciting and novel mechanism

of creating such phases using periodically driven external forces.

Authors present a full analysis of why such phases may appear as steady states in a system with a very long lifetime. This is the paper's main result and, potentially, it may deserve to be published in Nature Communication. For this, I suggest that the authors present arguments explaining

(i) why the mean-field approach is working in such driven systems and

(ii) that the states are stable against quantum fluctuations, provided that they appeared in equilibrium systems? This is an essential question for understanding of quantum field theory for low energy excitation around such ground states.

Reviewer #2 (Remarks to the Author):

The authors propose a pathway to an interesting correlated electronic phase in a driven lightly doped semiconductor. In contrast to most of the Floquet engineering, which deals with drive effects on noninteracting band structures, here the interaction and relaxation processes are included in order to determine stability of a trial nematic electronic state.

The key to stability is the exotic Mexican hat band structure that obtains in the rotating frame. Interactions are shown to favor carriers aggregating in one region of the Mexican hat, as opposed to be spread uniformly around it.

Theoretically, there are several nontrivial elements to the treatment, perhaps the most novel one is the self-consistent Floquet. Just like the drive, the order parameter is time-dependent with the same frequency, and hence does not invalidate single frequency Floquet treatment.

I find the set up quite appealing, and results well-discussed and physically plausible. The paper is high level of creativity and quality and as such satisfies criteria of Nature Communications.

I have a couple of comments and perhaps suggestions for the future work.

First, I found the title a bit misleading. From the title, it sounded like multiple liquid crystals would be offered; however, only nematic is discussed. The qualifier "gyro" appears only in the title. I think calling it what it is -- Floquet nematic -- would be more accurate, and not less exciting.

Second, it would be nice if authors could comment on possibility of other phases that have been discussed for this type of band structure. For instance, in Phys. Rev. Lett. 111, 185304 (2013) a proposal has been made for quasicrystalline states, where particles clump in multiple puddles around the band structure trench, if the interaction has a characteristic length scale. Screened Coulomb could be potentially sufficient for that.

Finally, it would be great if authors could comment on the (im)possibility of quasiperiodic states in time; that is, when the order parameter rotates at a frequency different than the drive frequency.

Reviewer #3 (Remarks to the Author):

The authors propose a new nonequilibrium phase with a spontaneous symmetry breaking via so-called Floquet engineering approach.

The model they consider is a (single) massive Dirac cone with broken time reversal symmetry. It can be seen as an effective model for one valley of a transition metal dichalcogenide, although for a realistic problem one has to consider a pair of valleys.

The idea to realize an instability toward spontaneously symmetry broken state is quite intuitive. One can introduce a Mexican-hat formed quasienergy bands by shifting the lower band by the photon energy upward and making an anticrossing of two bands. The obtained Floquet bands have a divergence in the density of states leading to the instability.

While the relaxation toward the "ground state" of the effective static problem is highly nontrivial in the Floquet setup, the authors seriously examine the existence of the symmetry-broken phase as a nonequilibrium steady state, using a kinetic equation approach for the distribution function of the Floquet states, combined with the mean-field approximation.

The authors provide careful and detailed analysis on this problem of the nonequilibrium distribution, and discuss the possibility of the spontaneous symmetry breaking in the periodically-driven setup. While the heating effect has been a central problem on the Floquet engineering of many-body states, I think that the analysis given in the present work should be useful for its future development.

I recommend to accept this manuscript after a minor revision.

I have a few questions/comments on the manuscript below:

1.

While the authors have shown that the Floquet ansatz with broken symmetry exists, its stability against perturbations has been unclear. Is it possible to check the absence of collective excitations with negative or imaginary energy?

Also, are there no heating processes via (nearly) gapless collective modes?

2.

I naively expect that the steady state with the rotating order parameter has a large energy dissipation rate. Is it possible to estimate how much energy is transferred to phonons/photons per unit time in the steady state, in order to check the experimental feasibility?

3. The driving intensity V must be large enough to have a significant Floquet gap (it must be larger than the feasible bath temperature, for instance), while a fatal heating effect occurs if it is too large.

I would like to know typical upper and lower bounds where the both constraints are satisfied (or some intensity dependence of the simulation).

4. It would be more instructive to put section numbers when refer to SM in the main text.

RESPONSE TO THE COMMENTS OF REFEREE 1

We thank the referee for his/her careful reading of the manuscript, and helpful questions and comments. We were happy to read the referee's positive assessment of our work. Below is our response to the referee's questions and comments.

1. "(i) why the mean-field approach is working in such driven systems"

We thank the referee for this insightful question. We argue that despite the non-equilibrium nature of the system we study, the application of a mean-field treatment of interactions can be justified at a similar level as for an equilibrium system. To this end, we identify a limit in which the system we study maps to an equilibrium system. This limit is realized when the gap of the semiconductor, and hence the driving frequency, is large compared to the Rabi frequency (driving amplitude). In this situation, and for the moment neglecting electron-hole radiative recombination, one can apply the rotating wave approximation to the full many-body dynamics including the system-bath coupling. In this regime, in the rotating frame the system maps to an equilibrium problem with a static Hamiltonian describing interacting electrons in a new bandstructure with modified system-bath couplings [V. M. Galitskii *et al.*, Zh. Eksp. Teor. Fiz. **57**, 207 (1969), K. I. Seetharam *et al.*, Phys. Rev. X **5**, 041050 (2015)]. Therefore, the steady state of the system would be a Gibbs state with respect to this static Hamiltonian. In the rotating frame, the mean field approach can be used to study the properties of this Gibbs state and map out a phase diagram with the same level of justification as in equilibrium.

Away from the above limit, the many-body dynamics involves additional processes which are not present in equilibrium and which lead to a deviation from the exact Gibbs state discussed above. These processes are often called Floquet-Umklapp processes, since the total initial quasi-energy of the electrons and excitations in the environment differs from the final value by an integer multiple of the driving frequency. The rates for these processes are suppressed in powers of the ratio of the driving amplitude to the driving frequency (in addition, radiative recombination is naturally a slow process relative to electron-phonon and electron-electron scattering), see Discussion and Section III of the SM. Therefore, in the system we study, the rates for Floquet-Umklapp processes are small compared with the rates for processes that relax the system to the Gibbs state exhibited by the system in their absence. Our expectation is that the steady state and the order parameter that it exhibits evolve smoothly as the rates of these processes are increased from zero. Therefore, the mean-field approach remains a good approach for studying symmetry breaking even when Floquet-Umklapp processes are present in the regime of interest.

We added a discussion about the justification for the mean-field treatment in the section where it is introduced.

2. "...(ii) that the states are stable against quantum fluctuations, provided that they appeared in equilibrium systems? This is an essential question for understanding of quantum field theory for low energy excitation around such ground states."

Indeed, for a static system spontaneous breaking of a continuous $U(1)$ rotation symmetry is expected to be accompanied by quantum fluctuations due to gapless Goldstone modes. Importantly, in any system with an underlying lattice structure the perfect continuous rotational symmetry will be broken down to a discrete rotation symmetry. Even when the rotational symmetry breaking (e.g., warping) terms that reduce the symmetry of the band structure of the static system are weak, at energies of interest these terms serve as a relevant perturbation which will pin the order parameter. This opens a gap for the Goldstone mode.

The role of quantum fluctuations of the order parameter in the type of driven systems that we study is an interesting open question that we intend to address in future work. In equilibrium, an analysis of quantum fluctuations can be performed using time dependent Hartree-Fock. An adaptation of such an analysis to non-equilibrium steady states is yet to be established. One possible approach is to allow long-wavelength, low frequency fluctuations of the order parameter, thus generalizing the self-consistent treatment of coupled kinetic and Floquet-Hartree-Fock equations which we developed in this work. In any case, since any physical realization of the physics we discuss in this paper requires an underlying lattice structure, the argument above implies that quantum fluctuations of the order parameter should be suppressed and therefore we expect the steady state to be stable.

The analysis in our paper consists of two components, numerical and analytical. While the analytical approach assumed $U(1)$ symmetry, the numerical simulations include the effect of the lattice. The excellent agreement between these two approaches shows that the rotational symmetry breaking due to the lattice does not qualitatively affect the mean-field phase diagram. However, as discussed above, deep in the ordered phase the rotational symmetry breaking would have an important role in suppressing the long-wavelength fluctuations of the order parameter.

In the revised version of the manuscript, we added a new paragraph to the "discussion" section discussing the role of rotational symmetry breaking in suppressing long-wave length fluctuations of the order parameter.

RESPONSE TO THE COMMENTS OF REFEREE 2

We thank the referee for his/her careful reading of the manuscript, and helpful comments and suggestions. We were happy to read the referee's positive assessment of our work. Below is our response to the referee's questions and comments.

1. *"First, I found the title a bit misleading. From the title, it sounded like multiple liquid crystals would be offered; however, only nematic is discussed. The qualifier "gyro" appears only in the title. I think calling it what it is -- Floquet nematic -- would be more accurate, and not less exciting."*

We thank the referee for pointing out that we used the term "gyro" only once, without elaborating on its significance. We also agree with the referee that we consider only a single type of liquid phase – a rotating ferromagnetic nematic liquid crystal phase. We thus propose to rename the manuscript to "Electronic Floquet Gyro-Liquid Crystal," singular. At the same time, we have added an explicit discussion explaining the significance of the prefix "gyro." In the revised manuscript we also refer to the type of order we found as "gyro-ferromagnetic nematic," which we believe most faithfully describes the phase. It emphasizes the main unique feature of the phase, which is a time-rotating magnetic order.

2. *"...it would be nice if authors could comment on possibility of other phases that have been discussed for this type of band structure. For instance, in Phys. Rev. Lett. 111, 185304 (2013) a proposal has been made for quasicrystalline states, where particles clump in multiple puddles around the band structure trench, if the interaction has a characteristic length scale. Screened Coulomb could be potentially sufficient for that. Finally, it would be great if authors could comment on the (im)possibility of quasiperiodic states in time; that is, when the order parameter rotates at a frequency different than the drive frequency."*

The possibility of having phases with various spatial and temporal orders is a fascinating direction for future research. One possibility is to consider long-ranged interactions, such as Coulomb or dipole interactions, which may lead to complex spatial orders, e.g., similar to those presented in *Phys. Rev. Lett.* **111**, 185304 (2013) and *Phys. Rev. B* **85**, 035116 (2012). In addition, the order parameter may depend on the polarization of the driving field. For example, circularly polarized light preserves the U(1) symmetry, leading to a uniform gap of the Floquet spectrum, yet, with an interesting pseudospin texture at the resonance ring. The pseudospins in the resonance ring may wind zero or two times, depending on the handedness of the drive. In the latter case, each two opposite points on the resonance ring correspond to the same pseudospin direction. Therefore, the spontaneous breaking of the spin-rotational symmetry leads to two Fermi pockets of opposite momenta on the resonance ring (see the discussion section of the main text).

We added a short discussion on the potential of achieving different types of orders and refer to the above papers in the Discussion section of the paper.

The possibility for obtaining a “time-crystal”, i.e., a steady state in which the order parameter oscillates with a different periodicity than the drive, and even exhibits quasi-periodic temporal order, is a very interesting direction. We believe that in principle it should be possible to stabilize a steady state in which the order parameter oscillates with a period which is equal to an integer multiple of the driving period. Similar phases have been discussed theoretically and experimentally [cf., arXiv:2012.08885, arXiv:1807.09884]. We do not know of any fundamental reason why this should be impossible in a driven-dissipative system. We intend to investigate the possibility of obtaining such phenomena in a driven solid state system in future work. We added a discussion regarding the possibility of obtaining these phenomena to the discussion section.

RESPONSE TO THE COMMENTS OF REFEREE 3

We thank the referee for his/her careful reading of the manuscript, and helpful questions and comments. We were happy to read the referee's positive assessment of our work. Below is our response to the referee's questions and comments.

1. *“While the authors have shown that the Floquet ansatz with broken symmetry exists, its stability against perturbations has been unclear. Is it possible to check the absence of collective excitations with negative or imaginary energy? Also, are there no heating processes via (nearly) gapless collective modes?”*

Our analysis is performed in the limit where the parameter κ , describing the competition between the heating and cooling processes, is small. In this regime, the system is always close to the limit described in our response to Referee 1, where the rotating wave approximation is valid and the system tends to a low-temperature equilibrium-like state in the rotating frame (i.e., in terms of the quasienergy bands). Precisely in this limit, i.e., when $\kappa = 0$, the steady state exactly maps to a zero-temperature equilibrium state. This mapping leads to a stable steady-state with no modes with imaginary frequency.

When κ is finite, yet small, we do not expect any mode to develop a significant imaginary frequency. In fact, the steady state may still be approximately mapped to a finite-temperature equilibrium state for each of the Floquet bands (cf. Fig. 3c in the main text and Section III of the supplementary material).

With regard to the possibility of gapless collective modes, we note that, due to crystalline symmetry of the underlying material, we expect the perfect U(1) rotational symmetry to be broken down to a lower, discrete symmetry. Even when the rotational symmetry breaking (e.g., warping) terms that reduce the symmetry of the band structure of the static system are weak, at energies of interest these terms serve as a relevant perturbation which will pin the order parameter and produce a gap for the collective modes. While the analytical component of our work assumed U(1) symmetry, the numerical simulations include the effect of the lattice. The excellent agreement between these two approaches shows that the rotational symmetry breaking due to the lattice does not qualitatively affect the mean-field phase diagram.

A full analysis requires developing a formalism which allows to treat collective modes of the system, for example employing time-dependent Hartree-Fock theory combined with the kinetic equation approach, and is beyond the scope of this paper. This is a very interesting direction for future work.

In the revised version of the manuscript, we added a new paragraph to the “discussion” section discussing future extensions of this work to incorporate long-wavelength fluctuations of the order parameter around the steady state.

2. *“I naively expect that the steady state with the rotating order parameter has a large energy dissipation rate. Is it possible to estimate how much energy is transferred to phonons/photons per unit time in the steady state, in order to check the experimental feasibility?”*

The steady state in our system corresponds to a current of particles first excited from the lower to the upper Floquet band via Floquet-Umklapp processes and then relaxed back to the lower band by phonon-assisted relaxation. In a single cycle of such process, each electron absorbs energy of $\hbar\Omega$. Assuming that the Floquet-Umklapp processes result mostly from the electron-photon interactions, we estimate the particle flux due to Floquet-Umklapp processes by $\dot{n} = \Gamma_{\text{ph}}$ [see above Eq.(6)]. Therefore, the total energy flux density dissipated in the process is given by $W = \dot{n}\hbar\Omega$. The total energy flux may be separated into the energy dissipated by photons and the rest, which is dissipated by phonons. We estimate the energy flux dissipated by photons by $W_\ell = \dot{n}E_g$, where E_g is the band gap of the semiconductor. Subsequently, the energy dissipated by phonons is given by $W_s = \dot{n}(\hbar\Omega - E_g)$. For a typical semiconductor of size 5×5 microns this yields $W_\ell \approx 4\text{mW}$ and $W_s \approx 60 \mu\text{W}$. The effective temperature of the electrons in the upper band can be estimated by Eq. (S87) in the supplementary material. For typical parameters and excitation density of $n_e \approx 10^{10} \text{cm}^{-2}$, we obtain an electron effective temperature of $T_e \sim 10 \text{K}$. Therefore, the cryogenic system used for such an experiment does not need to cool to a significantly lower temperature than this. Cryogenic systems with sufficient cooling power at this temperature are available and widely used.

In the revised version of the manuscript, we added an estimation of the heat dissipated in the system to the discussion section.

3. *“The driving intensity V must be large enough to have a significant Floquet gap (it must be larger than the feasible bath temperature, for instance), while a fatal heating effect occurs if it is too large. I would like to know typical upper and lower bounds where the both constraints are satisfied (or some intensity dependence of the simulation).”*

We focus on the controlled regime, where the system can be described by Eq. (1) and the parameters satisfy the conditions appearing below Eq. (1). An important condition for being in this regime is that the driving amplitude is much smaller than the frequency, $V \ll \hbar\Omega$. Under these conditions, Floquet-Umklapp processes arising from electron-phonon and electron-electron scattering are suppressed. Thus, in this regime, and for the interaction parameters not too large (see below), photon-assisted processes (i.e., radiative recombination) dominate the heating rate.

We will formulate our answer in terms of bounds on the Floquet gap, and afterwards relate this to the drive intensity. Our analysis relies on the Floquet states providing a good description for the steady state. The Floquet states significantly differ from the static Bloch states of the system (in the absence of the drive) only near the resonance, i.e., near the bottom and top of the upper and lower Floquet bands, respectively. In order for the steady state to be well described by populations of the Floquet states, and for the Floquet states even to be well-resolved, the Floquet gap must be larger than the inverse lifetimes of these states. We estimate the lower bound on the Floquet gap as the value at which it equals the scattering rate, $\Delta_{\text{min}} = \hbar/\tau_{\text{scat}}$. Below this

value, the diagonal ensemble of Floquet states is not an appropriate fixed point for the steady state distribution.

We now estimate the scattering rate for electrons at the bottom of the upper Floquet band. For these electrons, the main scattering process is phonon-assisted inter-Floquet-band relaxation. From Eq. (6) of the main text under steady state conditions, we estimate $1/\tau_{scat} \approx \Lambda_{inter} n_h = \Gamma_\ell/n_e$. Employing the expression for the excitation density n_e at half-filling [see Eq. (7)] we find $1/\tau_{scat} \approx \sqrt{\Gamma_\ell \Lambda_{inter}}$. Here, as in the main text, Γ_ℓ and Λ_{inter} capture radiative recombination and photon-assisted inter-Floquet-band relaxation, respectively.

Expressions for Γ_ℓ and Λ_{inter} as functions of the system's parameters are given in Eqs. (S56) and (S65) in the supplementary material. The flux Γ_ℓ is approximately independent of the drive intensity, while Λ_{inter} scales as $1/V$ in the limit $\Delta_F \gg 2k_R v_s$, where k_R is the resonance momentum and v_s is the speed of sound. Therefore, we estimate $\Gamma_\ell \approx A_R/\tau_\ell$, where τ_ℓ is the radiative recombination lifetime and $A_R = \pi k_R^2$. We furthermore estimate $\Lambda_{inter} \approx (\Delta_F \tau_\Lambda^2 A_{BZ})^{-1}$, with $\tau_\Lambda = (12\pi^3 g_s^2 \rho_s^0 k_R^2)^{-\frac{1}{2}}$. Here g_s and ρ_s^0 are the electron-phonon coupling and scale factor of the phonon density of states, as defined on page 8 of the main text (see also Sec. VII of the Supplementary Information). We then employ the relation between the Floquet gap and the driving intensity appearing in the last paragraph of the “model system and problem setup” section to obtain the lower bound for the intensity:

$$\Delta_{\min} = \left(\frac{\Gamma_\ell}{A_{BZ} \tau_\Lambda^2} \right)^{\frac{1}{3}},$$

in the units where $\hbar = 1$.

The upper bound on the Floquet gap can be estimated as the value at which the heating processes are dominated by the electron-electron interactions. Beyond this value of the Floquet gap, the heating rate rapidly increases with driving amplitude. This condition corresponds to $\Gamma_{ee} = \Gamma_\ell$. An expression for the rate Γ_{ee} appears in Eq. (S62) of the Supplementary Information. The upper bound for the Floquet gap reads

$$\Delta_{\max} = \left(\frac{8\pi^4 \Gamma_\ell \lambda_0^2}{A_R m_* U^2} \right)^{\frac{1}{2}},$$

where m_* is the effective mass of the semiconductor in the absence of the drive, and λ_0 is derived from the spin orbit coupling, see Eq. (1) of the main text.

For typical semiconductor parameters used in the manuscript and for normalized interaction strength $\tilde{U} \approx 3$ [see below Eq. (10) for definition of \tilde{U}], we obtain $\Delta_{\min} \approx 0.2$ meV and $\Delta_{\max} \approx 0.2$ eV.

We now obtain the amplitudes and the intensities of the applied electric fields corresponding to the limits above, for the case of driving by circularly polarized light at normal incidence to the 2D electron system. We use the formula relating the Floquet gap to the electric field amplitude of the driving field that appears in N. H. Lindner, G. Refael, and V. Galitski, Nat. Phys. **7**, 490 (2011), [Eq. (20) therein]. This yields $\mathcal{E}_{\min} \approx 4 \times 10^4$ V/m and $\mathcal{E}_{\max} \approx 4 \times 10^7$ V/m. The corresponding intensities read $\mathcal{I}_{\min} \approx 2 \times 10^6$ W/m² and $\mathcal{I}_{\max} \approx 2 \times 10^{12}$ W/m². The intensities used in recent experiments on Floquet engineered materials are comparable to the upper limit of this

range [for example, McIver et al., Nature Physics **16**, 38 (2020)]. Note that compared to these experiments, in the setup that we propose heating is reduced by working in a more favorable frequency regime. Specifically, we consider a frequency comparable to the band gap of a wide-gap semiconductor (rather than working at much lower mid-infrared frequencies). For a fixed value of the Floquet gap, Floquet-Umklapp scattering due to electron-electron and electron-phonon interactions is suppressed at large frequencies.

In the revised version of the manuscript, we added a discussion in which we estimate these upper and lower bounds of the driving intensity. We furthermore added a section to the SM in which we carefully derive these estimates and specify the values of all parameters used.

4. *“It would be more instructive to put section numbers when refer to SM in the main text.”*

In the new version of the manuscript, we added section numbers when refer to SM.

LIST OF CHANGES

1. We added a discussion describing possible phases in the presence of long ranged interactions in the “Discussion” section.
2. We added a discussion describing the pinning effect of the lattice to the “Discussion” section.
3. We added a new paragraph with an estimate of heating rate dissipated to the heat baths to the “Discussion” section.
4. We added a new paragraph with an estimation of the lower and upper bounds of the driving intensity to the “Discussion” section.
5. We modified our estimate of Λ_{inter} in the “Discussion” section.
6. We added section numbers in the references to Supplementary Material throughout the entire manuscript.
7. We added a discussion about the justification for the mean-field treatment, in the section where the mean field is introduced.
8. We added a new section in the supplementary material, which includes estimates for a typical experimental setup.
9. We changed the name of the symmetry broken phase to “gyro-ferromagnetic nematic” (GFN) phase throughout the manuscript.
10. We changed the title of the manuscript to “Electronic Floquet Gyro-Liquid Crystal”.
11. The affiliation of one of the authors has been modified.

REVIEWERS' COMMENTS

Reviewer #1 (Remarks to the Author):

The present report is the second review of the manuscript entitled "Electronic Floquet Gyro-Liquid Crystal." The first review addressed the noteworthiness and significance of the results. I have read the new version of the manuscript and the authors' responses to the Referee comments. I found that the authors considered my comments thoroughly and introduced changes to the manuscript where needed. These additions considerably improved the manuscript, which I recommend for publication in Nature Communications in the present form.

Reviewer #2 (Remarks to the Author):

I find that the authors addressed my comments and I recommend the paper for publication.

Reviewer #3 (Remarks to the Author):

I thank the authors for the additional clarifications.

In particular, the relationship between this work and the possible experimental realization becomes clearer.

I recommend publication of the paper in its current form.